# Evaluation of the Hematological and Serum Biochemistry Parameters in the Pre-Symptomatic and Symptomatic Stages of ALS Disease to Support Early Diagnosis and Prognosis

**DOI:** 10.3390/cells11223569

**Published:** 2022-11-11

**Authors:** Duygu Aydemir, Selcuk Surucu, Ayse Nazli Basak, Nuriye Nuray Ulusu

**Affiliations:** 1Department of Medical Biochemistry, School of Medicine, Koc University, Sariyer, Istanbul 34450, Turkey; 2Koç University Research Center for Translational Medicine (KUTTAM), Sariyer, Istanbul 34450, Turkey; 3Department of Anatomy, School of Medicine, Koc University, Sariyer, Istanbul 34450, Turkey; 4Suna and İnan Kıraç Foundation, Neurodegeneration Research Laboratory, NDAL-KUTTAM, School of Medicine, Koç University, Istanbul 34010, Turkey

**Keywords:** ALS, serum biochemistry, hematological parameters, early diagnosis, pre-symptomatic, symptomatic

## Abstract

Amyotrophic lateral sclerosis (ALS) is the most common motor neuron disease. Since there are no pathognomonic tests for ALS prognoses; clinical diagnoses of the disease take time and are usually difficult. Prognostic biomarkers are urgently needed for rapid and effective ALS prognoses. Male albino rats were divided into ten groups based on age: 0 (40–45 days old), A (70–75 days old), B (90–95 days old), C (110–115 days old), and D (130–135 days old). Each group was divided into two subgroups according to its mutation status: wild type (SOD1^WT^) or mutated (SOD1^G93A^). Serum biochemistry and hematological parameters were measured in 90 rats to evaluate possible biomarkers for faster ALS diagnoses and prognoses. Weight loss, cholesterol, creatinine, glucose, total bilirubin (TBIL), blood urine nitrogen (BUN), c-peptide, glucagon, PYY, white blood cell (WBC), lymphocyte (LYM), monocyte (MID), granulocyte (GRAN), red cell distribution width with standard deviation (RDW-SD), red cell distribution width with the coefficient of variation (RDW-CV), platelet (PLT), mean platelet volume (MPV), platelet distribution width (PDW), and procalcitonin (PCT) levels were changed in the SOD1^G93A^ rats compared to the SOD1^WT^ rats independently from aging. For the first time in the literature, we showed promising hematological and serum biochemistry parameters in the pre-symptomatic and symptomatic stages of ALS by eliminating the effects of aging. Our results can be used for early diagnoses and prognoses of ALS, improving the quality of life and survival time of ALS patients.

## 1. Introduction

Amyotrophic lateral sclerosis (ALS) is the most frequent adult-onset motor neuron disease affecting 1.7–2.1 per 100,000 people per year worldwide. A total of 90–95% of ALS cases are sporadic (sALS), whereas 5–10% of patients have familial ALS (fALS) [1]. Since there are no pathognomonic tests for the diagnosis and prognosis of ALS, the clinical diagnosis of the disease takes time and is usually difficult. Additionally, the presentation of symptoms can vary between patients in terms of disease progression. However, the diagnosis should be as rapid and effective as possible since the median survival is only 2–3 years, and approved drugs (riluzole and edaravone) can only slow the progression of the disease in the early stages. Since an ALS diagnosis takes 12–15 months, which is extremely late for the drugs to work effectively, diagnostic biomarkers are urgently required to diagnose ALS rapidly and effectively [2,3].

ALS causes a hypermetabolic state leading to malnutrition, a lower BMI, impaired blood metabolism, and decreased muscle mass in patients [4]. Hypermetabolic states can be evaluated using various serum biochemical biomarkers, including pre-albumin, cholesterol, albumin, creatinine, retinol-binding protein, transferrin, fibronectin, high-density lipid (HDL), and insulin-like growth factor [5]. Transferrin and albumin are indicators of insufficient protein intake over 15–20 days; on the other hand, pre-albumin, fibronectin, and retinol-binding protein are indicators of temporary nutrient deficiency [6]. Moreover, hematological parameters enable ALS diagnoses since blood is easily sampled and considered a potential biomarker source for ALS. Therefore, hematological and serum biochemistry biomarkers, for instance, neurofilaments, MCP-1, interleukins, and hemoglobin, can be evaluated as possible biomarkers, enabling the early diagnosis and prognosis of ALS [7,8].

In this study, we have evaluated various hematological and serum biochemistry biomarkers in SOD1^G93A^-mutated rats (SOD1^G93A^) expressing mutated human SOD1^G93A^ protein compared to wild-type rats (SOD1^WT^) for the early diagnosis and progression of the disease by eliminating the effects of aging for the first time in the literature.

## 2. Methods

### 2.1. Ethical Statement

All experimental procedures and animal use were approved by the Ethics Committee of Koc University with the number 2019.HADYEK.006. All procedures were performed by personnel trained in the techniques according to Koc University Animal Laboratory guidelines (Istanbul, Turkey). The facility operates according to the Guide for the Care and Use of Laboratory Animals, and the requirements of the Animal Welfare Act and Regulations as well as the Public Health Service Policy on Humane Care and Use of Laboratory Animals. All methods were performed in accordance with the relevant guidelines and regulations. All animals were euthanized by cervical dislocation following an overdose of isoflurane. Our report follows the recommendations in the ARRIVE (Animal Research: Reporting of In Vivo Experiments) guidelines. No human subjects were used during the experiments.

### 2.2. Animal Studies

Six male and six female SOD1^G93A^-mutated albino rats were purchased from Taconic with catalog number NTac: SD-Tg(SOD1G93A)L26H. The human mutant SOD1^G93A^ protein in the animals was 8-fold more endogenous in the pre-symptomatic stages and 16-fold more in the last stage of the disease (http://www.taconic.com/2148, accessed on 6 October 2022). Animals were inbred at the Animal Research Facility of Koc University, and 90 male rats weighing 140–650 g were used during the experiments. Isoflurane was used as an anesthetic, and cervical dislocation under anesthetic was performed to euthanize each animal. Isoflurane was used to avoid interfering with the biochemical measurements and minimize animal suffering.

The sample size was calculated via power analysis. Animals were housed as described previously in detail [9]. Animals were divided into ten groups based on age: 0 (40–45 days old), A (70–75 days old), B (90–95 days old), C (110–115 days old), and D (130–135 days old), and each group was divided into two subgroups according to their mutation status: SOD1^WT^ and SOD1^G93A^, respectively. Group C accounts for the early stage, and group D refers to the late stage of ALS, in which groups 0, A, and B represent the pre-symptomatic stages of ALS (Appendix A). All experimental procedures and animal use were approved by the Ethics Committee of Koc University with the number 2019.HADYEK.006.

### 2.3. Genotyping of SOD1^WT^ and SOD1^G93A^ Rats

Rat tails of each rat were collected, and 25 mg of tissue was used to isolate DNA via DNeasy Blood & Tissue Kit (Qiagen, Hilden, Germany). A total of 20 ng/uL of DNA was used to perform PCR testing with forward and reverse primers (Forward primer: 5′ GTG GCA TCA GCC CTA ATC CA 3′ and Reverse primer: 5′ CAC CAG TGT GCG GCC AAT GA 3′). Cycling conditions were as follows: denaturation at 95 °C for 1 min, extension at 95 °C for 15 s, annealing at 62 °C for 15 s and then at 72 °C for 20 s, and final extension at 72 °C for 8 min with 34 cycles. After PCR was performed, products were loaded into the 2% agarose gel to evaluate the corresponding bands. NDAL Laboratory conducted genotyping of SOD1^G93A^ and SOD1^WT^ rats at the Koc University Hospital.

### 2.4. Evaluation of the Hematological and Serum Biochemistry Parameters

Blood samples were taken directly from heart under anesthesia and collected into three tubes as additive-free vacutainers: BD Vacutainer^®^ Plastic Silicone Coated Interior (BD, Franklin Lakes, NJ, USA), BD Vacutainer^®^ Heparin Tubes (BD, Franklin Lakes, NJ, USA) and Vacusera mini 0.2 K3 EDTA tubes (Vacusera, İzmir, Turkey). Blood samples were collected in the Vacusera mini EDTA tubes to prevent coagulation and following blood collection, samples were immediately analyzed with Prokan PE-6800Vet to evaluate hematological parameters. Blood samples collected in the BD Vacutainer^®^ Plastic Silicone Coated Interior (BD, Franklin Lakes, NJ, USA) were allowed to clot for 15 min at room temperature and then centrifuged at 500× *g* for 15 min at 4 °C. After centrifugation, serum samples were collected from the upper phase and stored at −80 °C until ICP-MS analysis. Blood samples collected in the BD Vacutainer^®^ Heparin Tubes (BD, Franklin Lakes, NJ, USA) were centrifuges at 400× *g* for 25 min at 4 °C and serum in the upper phase was collected and stored at −80 °C until analysis. Samples were not thawed more than one time to avoid freeze–thaw cycles. Serum biochemistry parameters were evaluated via VetTest 8008 using VetTest serum biochemistry kits (General Health Profile).

### 2.5. Magpix Luminex Analysis

MAGPIX^®^ System of Luminex was used to evaluate hormone levels in the serum samples of SOD1^WT^ and SOD1^G93A^ rats via Milliplex kit (MERCK, Kenilworth, NJ, USA, # RMHMAG-84K). Blood samples were collected from rats and assembled in the Vacusera mini 0.2 K3 EDTA tubes (Vacusera, İzmir, Turkey). Samples were centrifuged at 400× *g* for 3 min and serum was collected and stored at −80 °C until analysis was performed. Serum samples were pipetted into a 96-well plate as duplicates, and the experimental procedure was followed according to the kit’s instructions.

### 2.6. Microwave Digestion of Serum Samples and Inductively Coupled Plasma Mass Spectrometry (ICP-MS)

A total of 100 µL of serum sample was digested in 10 mL of 65% SUPRAPURE^®^ nitric acid (HNO_3_). A microwave digestion system (Milestone START D) only equipped with the temperature control sensor was used to dissolve serum samples as described previously [10]. Following microwave digestion, samples were diluted at 1/20 in ultrapure water to be measured via ICP-MS. We evaluated Ag, Al, As, Ba, Be, Ca, Cd, Co, Cr, Cs, Cu, Fe, Ga, K, Li, Mg, Mn, Na, Ni, Pb, Rb, Se, Sr, Tl, U, V, and Zn levels in the SOD1^G93A^ and SOD1^WT^ rats. Trace and mineral element levels in the rat serum samples were evaluated using Agilent 7700x ICP-MS (Agilent Technologies Inc., Tokyo, Japan) as described previously [9,10].

### 2.7. Statistical Analysis

Graphpad Prism (9.0) software was used to analyze the data. Statistical analysis was performed by two-way ANOVA followed by multiple comparison. All data were represented as mean ± SD. MILLIPLEX^®^ Analyst 5.1 software was used to analyze raw data acquired by MILLIPLEX^®^ MAP kit # RMHMAG-84K.

## 3. Results

### 3.1. Animal Weight, Absolute Organ Weight, and Relative Organ Weights as % Body Weight

The animals were weighed just before euthanasia, and the bodyweight of SOD1^G93A^ rats decreased in groups C and D compared to the SOD1^WT^ rats (Figure 1). The relative organ weight (ROW) was calculated as a percentage of the total body weight (Table 1) and absolute organ weights are shown in Appendix A. The absolute organ weights did not change significantly between SOD1^G93A^ and SOD1^WT^ rats in any groups except the hearts in the groups D (Appendix A). The ROW of the liver, brain, heart, and kidney organs significantly increased in the SOD1^G93A^ rats of groups C and D compared to the SOD1^WT^ rats of the indicated groups. The ROW of the tissues and lungs increased in the SOD1^G93A^ rats of groups C and D compared to the SOD1^WT^ rats; however, this increase was only significant in group D between the WT and G93A rats. No significant changes have been observed between the splenic weights of SOD1^WT^ and SOD1^G93A^ rats belonging to any groups (Table 1).

### 3.2. Serum Biochemistry Parameters of SOD1^WT^ and SOD1^G93A^ Rats

The albumin, ALT, amylase, calcium, globulin, and TP levels in the serum did not significantly change between the SOD1^WT^ and SOD1^G93A^ rats (Table 2). The cholesterol levels increased in the SOD1^G93A^ rats compared to the SOD1^WT^ rats in each age group; however, this increase was only significant for groups 0, C, and D (Figure 2). The ALKP and creatinine levels decreased in the SOD1^G93A^ rats compared to the WT rats except for group 0; however, this decrease was only significant in groups C and D for creatinine (Figure 2). The TBIL levels increased dramatically in the SOD1^G93A^ rats belonging to groups C and D compared to the SOD1^WT^ rats (Figure 2). The BUN levels increased in the SOD1^G93A^ rats compared to the SOD1^WT^ rats in all groups except group B; however, this increase was only significant for group 0 (Figure 2).

### 3.3. Serum Hormone and Glucose Levels

The C-peptide levels increased in the ALS rats compared to the WT in groups 0, A, and B. In contrast, they significantly decreased in groups C and D. This decrease was almost double in the G93A rats compared with the WT rats (Figure 3). The glucagon levels of G93A rats have started to increase compared to the WT rats in group B. This trend has become significant in the groups C and D (Figure 3). The glucose levels in the SOD1^G93A^ rats increased compared to the SOD1^WT^ rats in groups A and B, whereas they decreased in the groups of 0, C, and D (Figure 3). The PYY levels decreased in the G93A rats compared to the WT in groups 0, A, and B, and increased in the groups C and D. On the other hand, the MCP-1 levels of the SOD1^G93A^ rats were less compared to those of the SOD1^WT^ rats in all groups (Figure 3).

### 3.4. Trace Element and Mineral Levels in the Serum Samples of SOD1^WT^ and SOD1^G93A^ Rats

The Na levels significantly increased in ALS rats compared to the WT rats in groups B, C, and D. The Mg levels decreased in the SOD1^G93A^ rats of groups A, B, and D, whereas they increased in groups 0 and C (Figure 4). The K levels increased in the SOD1^G93A^ of all groups compared to the SOD1^WT^ rats, but were only significant in groups 0, C, and D (Figure 4). The Fe levels significantly increased in the SOD1^G93A^ rats belonging to groups C and D and were decreased in group 0 compared to those of the SOD1^WT^ rats (Figure 4). The Zn levels were dramatically reduced in the SOD1^G93A^ rats of groups B, C, and D compared to those of the SOD1^WT^ rats (Figure 4).

### 3.5. Hematological Parameters of SOD1^WT^ and SOD1^G93A^ Rats

The percentage of Lymphocytes and the number of lymphocytes of SOD1^G93A^ rats decreased in all groups compared to those of the SOD1^WT^ rats; however, this decrease was significant in groups C and D. The percentage of monocytes increased in the SOD1^G93A^ rats compared to that of the SOD1^WT^ rats belonging to the groups B, C, and D. The number of monocytes and the number of granulocytes decreased in the SOD1^G93A^ rats compared to those of the SOD1^WT^ rats in all groups. The percentage of granulocytes increased in the SOD1^G93A^ rats compared to that of the SOD1^WT^ rats in all groups; however, this increase was significant in group C (Figure 5). The WBC and PLT levels decreased in the SOD1^G93A^ rats compared to the SOD1^WT^ in all groups. This decrease was significant in C and D. The HCT and MCV levels decreased in the SOD1^G93A^ rats compared to those of the SOD1^WT^ rats in groups C and D. The MCHC levels decreased in the SOD1^G93A^ rats compared to those of the SOD1^WT^ rats in groups 0, and were increased in groups C and D. The RDW-SD decreased in all groups except A for the SOD1^G93A^ rats compared to the SOD1^WT^ rats (Figure 5). The RDW-CV and PCT levels decreased in the SOD1^G93A^ rats in all groups except group A compared to those of SOD1^WT^ (Figure 5). We found that the RBC, HGB, and MCH levels decreased in the SOD1^G93A^ rats in groups 0, A, and D, and C increased in group D (Table 3).

## 4. Discussion

ALS patients require rapid and accurate diagnoses since the symptoms of the disease are not highly specific and can mimic the symptoms of other neurological disorders. Thus, developing prognostic and diagnostic biomarkers for ALS is urgently needed for effective and rapid diagnoses of the disease [11,12]. We have used SOD1G93A rats in this study since our transgenic rat model expresses human mutant SOD1^G93A^ protein. Furthermore, rats were chosen as the model because of the high blood volume and serum required to evaluate a wide range of biomarkers, including hematology, hormones, trace elements, minerals, and serum biochemistry.

Weight loss (WL) has been categorized as a clinical feature and predictive value for ALS and was observed in 56% to 62% of all cases [13]; it is associated with morbidity, mortality, and respiratory and functional loss in ALS. Thus, bodyweight management improves patients’ diagnoses [14]. The SOD1^G93A^ rats lost significantly more weight than the SOD1^WT^ rats in groups C and D, indicating that the WL probably started before the first symptoms showed (Figure 1). WL can result from different factors, including hypermetabolic state, decreased food intake, metabolic or hormonal levels, and physical activity status [15]. ROW is used to evaluate the toxicity of a substance and possible tissue damage in the organism [16]. According to our data, the ROW of all the SOD1^G93A^ rat tissues was significantly elevated compared to that of the SOD1^WT^ in groups C and D, except in the spleen (Table 1). Moreover, there was no significant difference in the absolute organ weights between SOD1^G93A^ and SOD1^WT^ rats except for the heart weight in group D (Appendix A). Thus, we propose that ALS disease progression in the early and late stages can induce impaired physiological homeostasis as well, since the ROWs of the liver, kidney, lung, brain, testis, and heart are altered (Table 1). Furthermore, the significant decrease in the heart tissue weight in the SOD1^G93A^ rats should be further evaluated and correlated with the muscle loss reduction (Appendix A). 

Biomarkers in the blood and serum can be used for the early and accurate diagnosis of the disease since riluzole and edaravone, the only two approved drugs, are only slightly effective in the early stages of ALS [2]. Since ALS causes a hypermetabolic state leading to malnutrition, weight, and muscle loss in patients, various biomarkers can be used to evaluate the diagnosis and progression of ALS [17]. We found that the albumin, TP, and Ca levels slightly increased in the SOD1^G93A^ rats compared to those of the SOD1^WT^ rats in groups B, C, and D, meaning that the changes have started at the pre-symptomatic stages (Table 2). The increased serum albumin and TP correlate with the body’s inflammatory status, and albumin levels positively correlate with ALS progression and survival. Additionally, albumin has antioxidant effects on metabolism [18,19]. Additionally, Ca homeostasis plays a vital role in ALS pathogenesis and disease progression, especially in SOD1-linked ALS; for instance, Ca buffering and metabolism are impaired in the motor neurons, mitochondria, and CNS even in the pre-symptomatic stages, according to various studies. Increased serum Ca levels in ALS patients have been found; however, possible mechanisms behind Ca homeostasis should be further investigated [20,21,22].

Creatinine has been characterized as an independent prognostic biomarker for ALS; it is elevated in the earlier phases of the disease as seen by the destruction in muscle mass and decreases in the late stages of the disease as a hallmark of malnutrition. Additionally, reduced creatinine levels are associated with reduced survival times and poor prognoses in male and female patients [23]. Furthermore, increased activities of SOD3, glutaredoxin 2 (GLRX2), and mitochondrial complex II have been found in the muscle tissue of ALS patients in the early stages, indicating an impaired energy metabolism in their muscles [24]. We have found that creatinine levels increased in the SOD1^G93A^ rats of group 0 and decreased compared to the SOD1^WT^ rats from aging and disease progression in groups A, B, C, and D (Figure 2), indicating that creatinine could be used as a diagnostic and prognostic biomarker in ALS.

Cholesterol is being discussed as another possible biomarker for ALS since neuron death occurs during the disease progression, resulting in elevated cholesterol levels in the CSF [25]. Interestingly, increased oxidative stress in the SOD1^G93A^ mice results in the formation of oxidized cholesterol products in the rats’ serum [26]. Elevated levels of cholesterol have been observed in ALS patients. Increased cholesterol levels are correlated with a risk of ALS and poor survival, resulting from oxidized cholesterol products and their harmful effects on metabolic pathways [27]. We found that the cholesterol levels in the mutated rats increased compared to those of the wild-type ones, and the increase was higher in groups C and D (Figure 2); thus, cholesterol can be used as a diagnostic biomarker for ALS.

ALKP is a liver enzyme responsible for breaking down proteins and transporting phosphate groups, and lower levels of ALKP are associated with malnutrition in humans [28]. We have shown that the ALKP levels started to decrease in the SOD1^G93A^ rats compared to those in the wild-type ones in group B; however, these changes were not significant (Figure 2). There have not been any studies that show the impact of ALKP as a biomarker in the pre-symptomatic and symptomatic stages of ALS; therefore, human studies with a high number of patients should be conducted to evaluate ALKP as a possible diagnostic biomarker in the future (Figure 2). Additionally, bilirubin is a potential inflammatory biomarker in inflammation-linked diseases, such as ALS, multiple sclerosis (MS), Alzheimer’s disease, and diabetes [29]. We found that the TBIL levels dramatically increased in the mutated rats compared to those in the wild-type ones in the symptomatic stages (C and D); thus, TBIL can be used as a possible biomarker for ALS diagnoses (Figure 2). Lastly, the BUN levels increased in the mutated rats compared to those of the wild type ones according to our data (Figure 2); however, the importance of BUN as a potential biomarker has not been studied until now, and thus a study should be conducted with a human cohort in the future.

ALS is a hypermetabolic disease; however, the possible mechanisms contributing to the energy metabolism homeostasis are unknown. Besides serum biochemistry biomarkers, we evaluated serum hormone and glucose levels in the SOD1^WT^ and SOD1^G93A^ rats (Figure 3). Glucose metabolism is considered one of the possible targeting approaches to cure people with ALS since it is vital for OXPHOS, the synthesis of neurotransmitters, and oxidative stress metabolism [30]. Glucagon induces the catabolism of the glycogen storages in the body, resulting in the release of glucose into the blood. Moreover, c-peptide can be measured instead of insulin since it is more stable and has a longer half-life. Additionally, it is produced in equal amounts with insulin, and both are responsible for lowering blood glucose [31]. Interestingly, we found that serum glucagon levels increased in the SOD1^G93A^ rats compared to those of the SOD1^WT^ rats in groups B, C, and D (Figure 3). Despite significantly increased glucagon levels, the blood glucose and c-peptide levels decreased in the AS rats compared to the WT ones in the symptomatic stages. Therefore, increased glucagon levels, despite reduced blood glucose levels, can be used as a diagnostic biomarker in ALS patients (Figure 3).

In addition, PYY is a gut hormone involved in energy expenditure, appetite, and fat oxidation. Increased levels of PYY in the blood result in increased energy expenditure, elevated levels of fat oxidation, and decreased appetite [32]. However, we have found that PYY levels decreased in the mutated rats compared to the wild-type ones in the pre-symptomatic stages but increased in the symptomatic stages (C and D) (Figure 3). MCP-1 expression is induced by insulin and plays a role in diabetes because of impaired glucose metabolism. Increased levels of MCP-1 are associated with impaired adipocyte functioning and decreased glucose uptake into the cells [33]. We have found that MCP-1 levels increased in the SOD1^G93A^ rats compared to those of the SOD1^WT^ rats in all groups; thus MCP-1 levels can be evaluated in humans as potential diagnostic biomarkers (Figure 3). All these data indicate that the increased catabolic activity and impairment in glucose metabolism started in the pre-symptomatic stages and worsened in the late stages of ALS. Thus, PYY, MCP-1, glucagon, c-peptide, and glucose levels can be evaluated as possible biomarkers for ALS diagnoses and progression as one of the hallmarks of impaired glucose metabolism [30].

We have also evaluated hematological parameters as potential biomarkers in the SOD1^WT^ and SOD1^G93A^ rats, which have become a hot topic over the past couple of years. RBC and MCH levels have been suggested as possible early biomarkers of the early stages of ALS by an in-silico study [34]. Additionally, decreased RBC counts have been reported as an early biomarker in ALS [35]. Further, increased HGB levels are associated with prognoses and reduced survival rates in ALS [36]. We found that RBC, HGB, and MCH levels decreased in the SOD1^G93A^ rats in groups 0, A, C, and D; however, they increased in group D (Table 3). We suggest that MCH, HGB, and RBC counts can be possible biomarkers in ALS and provide insight into the disease progression.

Hematological parameters indicating the impairment in the peripheral immune system and inflammation are vital for ALS prognoses and diagnoses since neuroinflammation contributes to ALS pathogeneses [37]. Increased monocyte, granulocyte, and neutrophil levels and decreased lymphocyte counts have previously been reported in ALS patients [37,38,39]. We found that monocyte and granulocyte counts started to increase in the mutated rats in the pre-symptomatic stages (B) and continued in the symptomatic stages (C and D) (Figure 4). In groups B, C, and D, the lymphocyte levels decreased in the SOD1^G93A^ rats (Figure 4). Therefore, granulocyte, lymphocyte, and monocyte levels can be used as potential biomarkers in ALS diagnoses and prognoses, indicating an impaired inflammation and immune response, which is one of the hallmarks of the disease [1,37]. In this study, we have reported that the MCH, RDW-SD, MCV, RDW-CV, PCT, and HCT values significantly decreased in the SOD1^G93A^ rats in comparison with those of the SOD1^WT^ rats in the early stage (C) and late stage (D) of ALS for the first time in the literature (Figure 4). We found that the WBC levels decreased in the mutated rats compared to the wild-type ones in all groups and the PLT levels decreased in the mutated rats in the pre-symptomatic (B) and symptomatic groups (C and D) for the first time in the literature as well (Figure 4). All the indicated parameters can be used as hallmarks of impaired blood homeostasis and immune systems since WBC, PLT, MCH, RDW-SD, HCT, MCV, PCT, and RDW-CV levels are impaired by the alteration of the indicated metabolisms [40]. Furthermore, these parameters should be evaluated in a human cohort to validate the accuracy of our data.

Serum mineral and trace element levels are vital for various metabolisms, such as antioxidant homeostasis, blood metabolism, neuron integrity, and nutrition balance; for instance, Fe plays a crucial role in the energy, blood, and oxidative stress metabolisms, and elevated levels of Fe indicate higher levels of oxidative stress in the organism. Additionally, elevated levels of iron can disturb Ca, Cu, and Zn levels in the body. We found that the Zn, Fe, Na, Mg, and K levels were impaired in the SOD1^G93A^ rats compared to those of the SOD1^WT^ rats, indicating impaired oxidative stress metabolism, neuronal homeostasis, and mineral metabolism [2,10,41,42,43,44,45] (Figure 5). On the other hand, ion channels, such as chloride channels in the muscle tissue, have been discussed as potential therapy options in several diseases, including ALS [46,47]. In ALS, muscle–nerve communication is impaired, associated with the loss of control of voluntary muscles and paralysis. For instance, sarcolemma ion channels, including Cl^−,^ K^+^, Na^+^, and Ca ^2+^ channels, regulate muscle function, plasticity, contraction, and muscle excitability [47]. Therefore, these minerals and trace elements can be evaluated as potential biomarkers for ALS diagnoses and prognoses.

In conclusion, for the first time in the literature, we have investigated a wide range of biomarkers in the blood and serum by eliminating the effects of aging and indicated promising biomarkers that enable us to diagnose ALS in its early stages.

## Figures and Tables

**Figure 1 cells-11-03569-f001:**
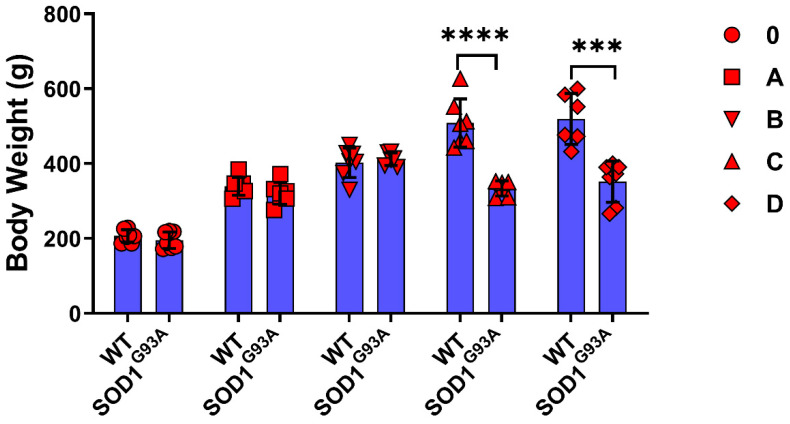
Body weights of the SOD1^G93A^ and SOD1^WT^ rats. All results were given as mean ± SD of *n* = 6–8 animals for each group. Notes: *** *p* ≤ 0.001, and **** *p* ≤ 0.0001.

**Figure 2 cells-11-03569-f002:**
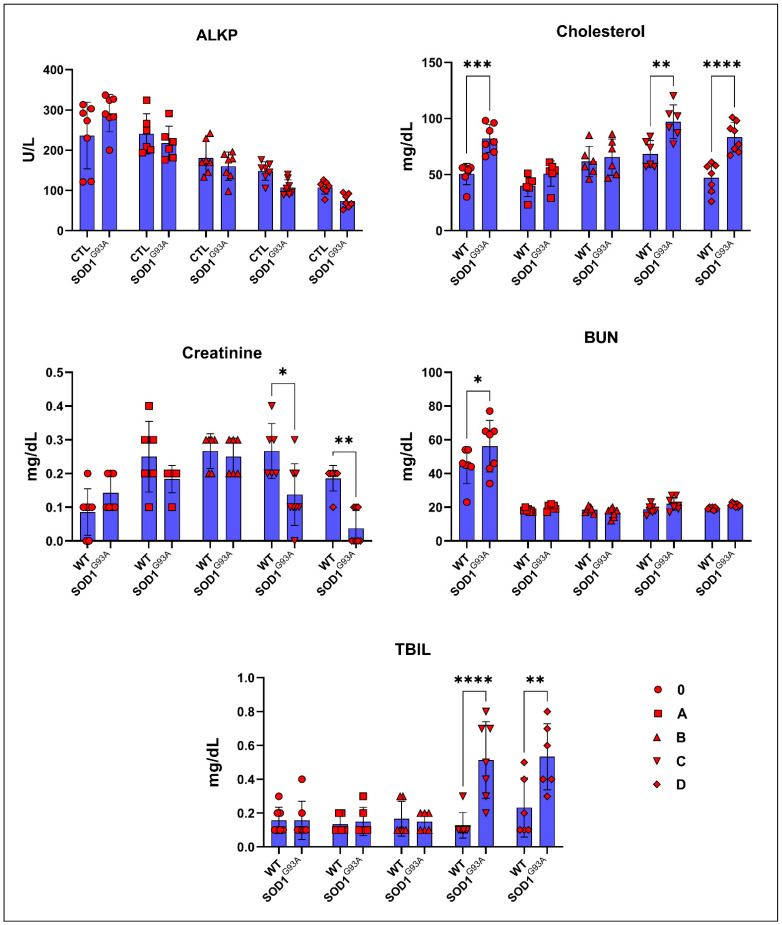
Serum biochemical parameters of the SOD1^G93A^ and SOD1^WT^ rats. All results were given as mean ± SD of *n* = 6–8 animals for each group. Notes: * *p* ≤ 0.05, ** *p* ≤ 0.01, *** *p* ≤ 0.001, and **** *p* ≤ 0.0001.

**Figure 3 cells-11-03569-f003:**
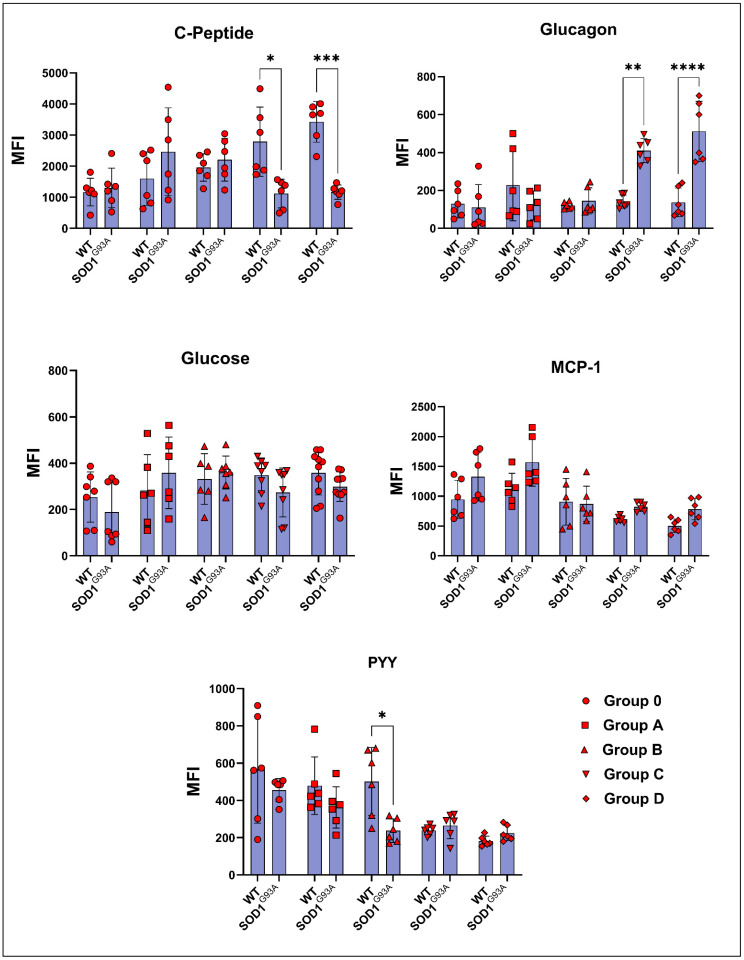
Serum hormone parameters of the SOD1^G93A^ and SOD1^WT^ rats. All results were given as mean ± SD of *n* = 6–8 animals for each group. Notes: * *p* ≤ 0.05, ** *p* ≤ 0.01, *** *p* ≤ 0.001, and **** *p* ≤ 0.0001.

**Figure 4 cells-11-03569-f004:**
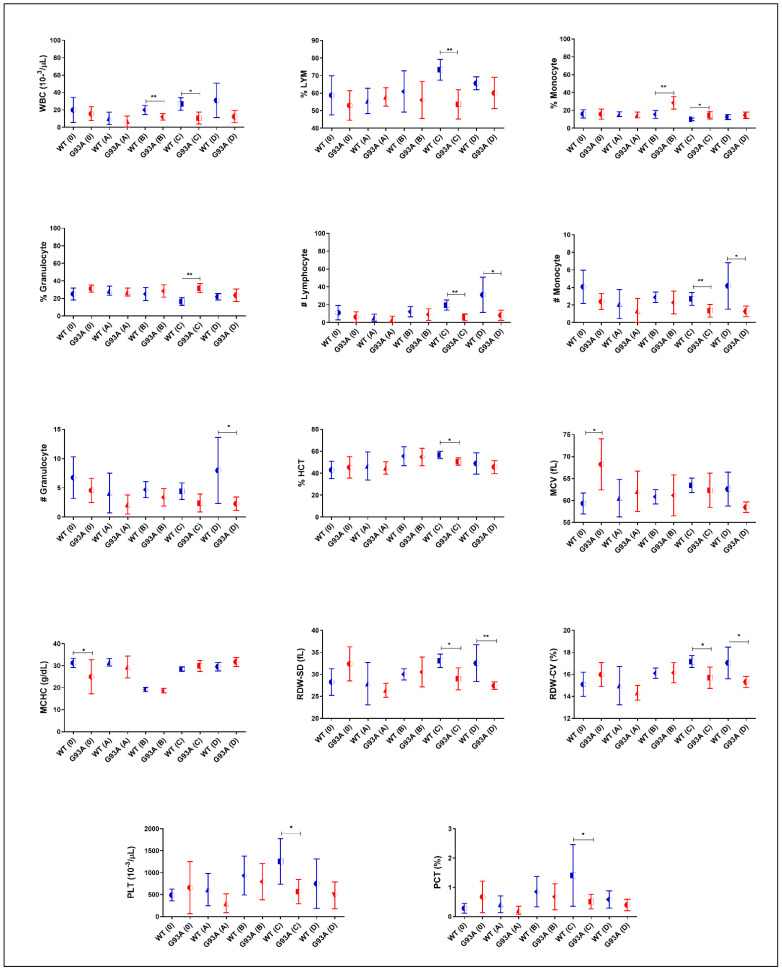
Blood parameters of the SOD1^G93A^ and SOD1^WT^ rats. All results were given as mean ± SD of *n* = 6–8 animals for each group. Notes: * *p* ≤ 0.05, ** *p* ≤ 0.01.

**Figure 5 cells-11-03569-f005:**
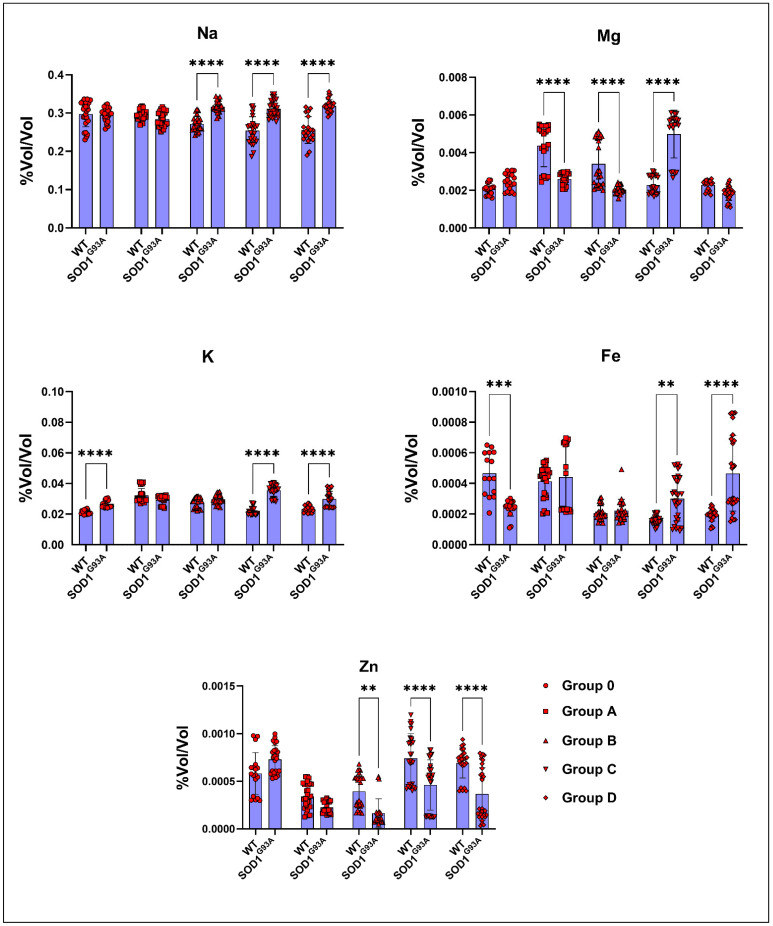
Serum mineral and trace element levels of the SOD1^G93A^ and SOD1^WT^ rats. All results were given as mean ± SD of *n* = 6–8 animals for each group. Notes: ** *p* ≤ 0.01, *** *p* ≤ 0.001, and **** *p* ≤ 0.0001.

**Table 1 cells-11-03569-t001:** Relative organ weight is represented by the percentage of body weight for each rat. All results were given as mean ± SD. ^a^ represents significantly different from SOD1^WT^ group (*p* ≤ 0.05), ^b^ represents significantly different from SOD1^WT^ group (*p* ≤ 0.001), and ^c^ represents significantly different from control SOD1^WT^ group (*p* ≤ 0.0001).

Serum Biochemistry Parameters								
		Brain	Lung	Liver	Spleen	Heart	Kidney	Testis
** *Group 0* **	** *WT* **	0.828 ± 0.119	0.609 ± 0.047	4.752 ± 0.875	0.236 ± 0.021	0.356 ± 0.035	0.832 ± 0.071	1.291± 0.082
	** *G93A* **	0.923 ± 0.164	0.660 ± 0.040	4.057 ± 0.595	0.246 ± 0.018	0.367 ± 0.033	0.856 ± 0.034	1.226 ± 0.029
** *Group A* **	** *WT* **	0.607 ± 0.045	0.505 ± 0.037	4.196 ± 0.443	0.203 ± 0.030	0.338 ± 0.016	0.803 ± 0.093	1.028 ± 0.063
	** *G93A* **	0.578 ± 0.084	0.476 ± 0.072	4.292 ± 0.595	0.198 ± 0.020	0.329 ± 0.013	0.761 ± 0.078	1.073 ± 0.100
** *Group B* **	** *WT* **	0.502 ± 0.041	0.449 ± 0.078	3.903 ± 0.194	0.171 ± 0.022	0.317 ± 0.026	0.724 ± 0.057	0.854 ± 0.069
	** *G93A* **	0.473 ± 0.053	0.486 ± 0.056	3.671 ± 0.427	0.189 ± 0.059	0.284 ± 0.031	0.688 ± 0.225	0.824 ± 0.054
** *Group C* **	** *WT* **	0.432 ± 0.061	0.526 ± 0.103	3.528 ± 0.195	0.156 ± 0.010	0.278 ± 0.014	0.657 ± 0.049	0.793 ± 0.111
	** *G93A* **	0.561 ± 0.075 ^b^	0.601 ± 0.088	3.977 ± 0.316 ^b^	0.164 ± 0.015	0.333 ± 0.021 ^c^	0.755 ± 0.049 ^c^	0.876 ± 0.128
** *Group D* **	** *WT* **	0.429 ± 0.042	0.518 ± 0.096	3.355 ± 0.273	0.163 ± 0.030	0.297 ± 0.021	0.632 ± 0.038	0.795 ± 0.101 ^b^
	** *G93A* **	0.657 ± 0.119 ^c^	0.660 ± 0.125 ^a^	3.886 ± 0.445 ^a^	0.164 ± 0.022	0.341 ± 0.026 ^c^	0.872 ± 0.084 ^c^	1.06 ± 0.217

**Table 2 cells-11-03569-t002:** Serum levels of albumin, alanine transaminase (ALT), amylase, calcium, globulin, and total protein (TP) in the SOD1^WT^ and SOD1^G93A^ rats, given as mean ± SD of *n* = 7 animals for each group.

Serum Biochemistry Parameters							
		Albumin (g/dL)	ALT (U/L)	Amylase (U/L)	Calcium (mg/dL)	Globulin (g/dL)	TP (g/dL)
** *Group 0* **	** *WT* **	3.11 ± 0.76	53.50 ± 15.33	1327 ± 256.8	10.39 ± 0.42	2.657 ± 0.67	5.743 ± 1.43
	** *G93A* **	2.80 ± 0.08	57.00 ± 6.30	1249 ± 298.9	9.957 ± 0.37	2.443 ± 0.20	5.214 ± 0.25
** *Group A* **	** *WT* **	2.80 ± 0.35	76.20 ± 24.16	1653 ± 390.4	10.72 ± 0.94	2.90 ± 0.45	5.700 ± 0.71
	** *G93A* **	2.61 ± 0.02	69.83 ± 10.03	1624 ± 284.7	10.06 ± 0.11	2.483 ± 0.18	5.100 ± 0.43
** *Group B* **	** *WT* **	2.71 ± 0.24	77.33 ± 12.33	1556 ± 129.4	10.08 ± 0.23	2.90 ± 0.12	5.617 ± 0.34
	** *G93A* **	3.17 ± 1.30	74.17 ± 19.76	1516 ± 302.1	10.23 ± 0.47	2.933 ± 0.47	5.667± 0.25
** *Group C* **	** *WT* **	2.80 ± 0.23	65.60 ± 13.83	1555 ± 136.6	9.980 ± 0.32	2.880 ± 0.30	5.680 ± 0.47
	** *G93A* **	2.98 ± 0.22	65.11 ± 8.22	1467 ± 213.1	10.40 ± 0.56	3,170 ± 0.23	6.060 ± 0.27
** *Group D* **	** *WT* **	2.86 ± 0.05	58.67 ± 19.35	1559 ± 227.2	10.00 ± 0.43	2.800 ± 0.43	5.583 ± 0.31
	** *G93A* **	3.02 ± 0.25	59.20 ± 13.75	1366 ± 387.4	10.35 ± 0.57	2.810 ± 0.35	5.820 ± 0.46

**Table 3 cells-11-03569-t003:** Hematological parameters in the SOD1^WT^ and SOD1^G93A^ rats were given as mean ± SD of *n* = 7 animals for each group. Note: Red blood cell (RBC), hemoglobin (HBG), mean corpuscular hemoglobin (MCH), mean platelet volume (MPV), platelet distribution width (PDW).

		Hematological Parameters				
		RBC (10^−6^/µL)	HGB (g/dL)	MCH (pg)	MPV (fL)	PDW (%)
** *Group 0* **	** *WT* **	7.024 ± 1.26	13.58 ± 3.03	19.10 ± 1.21	7.280 ± 0.28	8.320 ± 0.81
	** *G93A* **	6.528 ± 1.15	13.28 ± 2.73	16.77 ± 5.02	7.967 ± 0.70	9.083 ± 1.38
** *Group A* **	** *WT* **	7.640 ± 1.62	14.66 ± 3.77	18.99 ± 1.07	8.286 ± 1.48	8.743 ± 1.01
	** *G93A* **	7.475 ± 0.80	14.18 ± 2.03	17.64 ± 2.90	8.960 ± 1.89	8.620 ± 0.94
** *Group B* **	** *WT* **	9.156 ± 1.86	17.63 ± 3.51	19.11 ± 0.94	8.600 ± 1.42	8.829 ± 0.87
	** *G93A* **	9.109 ± 1.24	16.77 ± 2.96	18.61 ± 1.06	8.500 ± 1.62	8.914 ± 1.69
** *Group C* **	** *WT* **	8.926 ± 0.35	16.14 ± 0.83	18.02 ± 0.57	10.32 ± 2.89	10.42 ± 2.59
	** *G93A* **	8.416 ± 0.78	15.22 ± 2.21	17.96 ± 1.09	9.120 ± 1.11	8.720 ± 1.04
** *Group D* **	** *WT* **	7.803 ± 1.38	14.48 ± 3.03	29.53 ± 1.99	9.700 ± 2.67	9.967 ± 2.62
	** *G93A* **	8.151 ± 1.25	15.15 ± 3.05	31.69 ± 2.07	9.225 ± 1.58	8.788 ± 0.90

## Data Availability

Our data are not public since our study was conducted for the first time in the literature; however, the datasets used and analyzed during this study are available from the corresponding author upon reasonable request.

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
