# Peer review of "Evaluation of the Hematological and Serum Biochemistry Parameters in the Pre-Symptomatic and Symptomatic Stages of ALS Disease to Support Early Diagnosis and Prognosis"

_cells, 2022, doi:10.3390/cells11223569_

Round 1

Reviewer 1 Report

This manuscript describes an important aspect of ALS. To identify recognized biomarkers at the aim to diagnose and cure the pathology.

However, the most important points (the biomarkers having a significant weight) should be better explained and valorized, especially in the discussion.

The manuscript needs to be greatly improved.

Methods

It is not clear how and when blood have been collected

Results

Since body weight and the muscle weight is reduced during ALS one can expect a reduction of the other organs. It is possible to give an explanation?

Fig 2.

Please explain the acronyms. In this form it is difficult to follow.

Paragraph 3.4

It can be useful to measure also chloride element, other than calcium. It should be underlined that skeletal muscle in a chloride free solution is unstable leading to an increase of sarcolemma excitability with modification of its function.  For references the authors may use the following articles (Camerino et al. Sci Rep. 2019 Feb 28;9(1):3185. doi: 10.1038/s41598-019-39676-3) (Fracchiolla et al. J Med Chem. 2009 Oct 22;52(20):6382-93. doi: 10.1021/jm900941b).

Discussion

Edaravone is not so efficacy.

Please change the reference with numbers. i.e.  Fang et al;  Chiò et al.

Line 254

I think that the sentence “Increased Ca levels in ALS patients have been found” refers to a cytoplasmic Ca increase.

There are also other parameters that may have an importance also considering the severe skeletal muscle involvement (Lanznaster et al. Biomedicines. 2022 Jun 2;10(6):1307. doi: 10.3390/biomedicines10061307).

Finally, it should be introduced in the text the reason of the choose of the transgenic rat. The rat model can be close to human pathology? There are differences with respect to the mouse model?

Author Response

ANSWER TO CRITICS

Comment

This manuscript describes an important aspect of ALS. To identify recognized biomarkers at the aim to diagnose and cure the pathology. However, the most important points (the biomarkers having a significant weight) should be better explained and valorized, especially in the discussion. The manuscript needs to be greatly improved.

Answer

Dear Reviewer, thank you for your precious comments, we have revised and rewritten our manuscript according to your comments. All changes and additions were indicated in the red color in the manuscript.

Comments

It is not clear how and when blood have been collected.

Answer

We have rewritten 2.4. Evaluation of the hematological and serum biochemistry parameters part according to your comments in detailed as below.

“Blood samples were taken directly from heart under anesthesia and collected into three tubes as additive-free vacutainers BD Vacutainer® Plastic Silicone Coated Interior (BD, USA), BD Vacutainer® Heparin Tubes (BD, USA) and VACUSERA mini 0.2 K3EDTA tubes (VACUSERA, TR). Blood samples were collected in the VACUSERA mini EDTA tubes to prevent coagulation and following blood collection, samples were immediately analyzed with PROKAN PE-6800Vet to evaluate hematological parameters. Blood samples collected in the BD Vacutainer® Plastic Silicone Coated Interior (BD, USA) allowed to clot for 15 minutes at room temperature and then centrifuged at 500 x g for 15 min at 4 °C. After centrifugation, serum samples were collected from the upper phase and stored at – 80 °C until ICP-MS analysis. Blood samples collected in the BD Vacutainer® Heparin Tubes (BD, USA) were centrifuges at 400 g for 25 min at 4 °C and serum in the upper phase was collected to store at -80 °C until analysis. Samples were not thawed more than one time to avoid freeze-thaw cycles. Serum biochemistry parameters were evaluated via VetTest 8008 using VetTest serum biochemistry kits (General Health Profile).”

Comment

Since body weight and the muscle weight is reduced during ALS one can expect a reduction of the other organs. It is possible to give an explanation?

Answer

Thank you very much for your precious comments and contributions. Body weight, relative organ weight and organ weight should be evaluated together as we represented in our study. We did not show or measure muscle mass, however during sacrification we have seen muscle loss in the mutated rats compared to the wild-type ones. Main reason of the body weight loss is suggested muscle weight and fat loss during ALS (https://doi.org/10.1371/journal.pone.0251087, doi: 10.1212/WNL.0b013e3182840689), this can be reason of the reduced body weight but not the organ weight.

We have calculated absolute organ weights, but we just used relative organ weights in the previous version of the manuscript. We have prepared a supplementary figure (supplementary Figure.1.) to show absolute organ weights according to your comments. There was no significant change has been found in tissue weights between SOD1G93A and SOD1WT rats in all groups except heart tissue. However relative organ weights are the important for us because % body weight indicates differences between body mass index and organ weights shown Table.1. Also, we have revised discussion and results parts according to your comments as written below.

“On the other hand, absolute organ weights are represented in the Supplementary Fig.1. and there was no significant difference in the absolute organ weights between SOD1G93A and SOD1WT rats except heart weight in the group D (Supplementary Fig.1). Thus, we can propose that ALS disease progression in the early and late stages can induce impaired physiological homeostasis as well since ROW of the liver, kidney, lung, brain, testis, and heart has been altered (Table 1). On the other hand, significant decrease in the heart tissue in the SOD1G93A rats should be further evaluated correlated with the muscle loss reduction (Supplementary Fig.1).

Supplementary Figure.1. Absolute organ weight in SOD1G93A and SOD1WT

Comment

Fig 2. Please explain the acronyms. In this form it is difficult to follow.

Answer

We have rewritten legend of figure 2 as written below.

“Figure.2. Serum biochemical parameters including alkaline phosphatase (ALKP), cholesterol, creatinine, blood urea nitrogen (BUN) and total bilirubin (TBIL) of the SOD1G93A and SOD1WT rats were represented in the figure. All results were given as mean ± SD of n=6-8 animals for each group. Notes: * (p ≤ 0.05), ** (p ≤ 0.01), ***(p ≤ 0.001) and **** (p ≤ 0.0001)”

Comment

Paragraph 3.4

It can be useful to measure also chloride element, other than calcium. It should be underlined that skeletal muscle in a chloride free solution is unstable leading to an increase of sarcolemma excitability with modification of its function.  For references the authors may use the following articles (Camerino et al. Sci Rep. 2019 Feb 28;9(1):3185. doi: 10.1038/s41598-019-39676-3) (Fracchiolla et al. J Med Chem. 2009 Oct 22;52(20):6382-93. doi: 10.1021/jm900941b).

Answer

Dear Reviewer, thank you very much for your precious comments. We have evaluated Ag, Al, As, Ba, Be, Ca, Cd, Co, Cr, Cs, Cu, Fe, Ga, K, Li, Mg, Mn, Na, Ni, Pb, Rb, Se, Sr, Tl, U, V, and Zn concentrations in the serum samples of SOD1G93A and SOD1WT rats via optimized method by us (https://doi.org/10.1016/j.dib.2020.105218). We have added a detailed explanation in the manuscript as written below.

2.6. Microwave digestion of serum samples and inductively coupled plasma mass spectrometry (ICP-MS)

100 µl of serum sample was digested in 10 ml of 65% SUPRAPURE® nitric acid acid (HNO3). A microwave digestion system (Milestone START D) only equipped with the temperature control sensor was used to dissolve serum samples as described previously [10]. Following microwave digestion, samples were diluted at 1/20 in ultrapure water to be measured via ICP-MS. We have evaluated Ag, Al, As, Ba, Be, Ca, Cd, Co, Cr, Cs, Cu, Fe, Ga, K, Li, Mg, Mn, Na, Ni, Pb, Rb, Se, Sr, Tl, U, V, and Zn levels in the SOD1G93A and SOD1WT rats. Trace and mineral element levels in the rat serum samples were evaluated using Agilent 7700x ICP-MS (Agilent Technologies Inc., Tokyo, Japan) as described previously [9,10].”

Unfortunately, we did not measure chloride levels. Our study was done in 2020, even we would like to measure chloride we do not have serum samples currently. However, we have added new information in the manuscript from the references which you have suggested as written below.

“Serum mineral and trace element levels are vital for various metabolisms such as antioxidant homeostasis, blood metabolism, neuron integrity, and nutrition balance; for instance, Fe plays a crucial role in the energy, blood, and oxidative stress metabolisms and elevated levels of Fe indicate higher levels of oxidative stress in the organism. Additionally, elevated levels of iron can disturb Ca, Cu, and Zn levels in the body. We have found that Zn, Fe, Na, Mg, and K levels were impaired in the SOD1G93A rats compared to the SOD1WT rats, indicating impaired oxidative stress metabolism, neuronal homeostasis, and mineral metabolism  [2,10,41–45] (Fig. 5). On the other hand, ion channels such as chloride channels in the muscle tissue are discussed potential therapy options in several diseases including ALS [46,47]. In the ALS, muscle-nerve communication is the impaired associated with the lose control of voluntary muscle function and paralysis. For instance, sarcolemma ion channels including Cl−, K+, Na+ and Ca 2+ channels regulate muscle function, plasticity, contraction and muscle excitability [47]. Therefore, these minerals and trace elements can be evaluated as potential biomarkers for ALS diagnosis and prognosis.”

Comment

Edaravone is not so efficacy.

Answer

We have rewritten the first paragraph of the discussion as below according to your comments.

“ALS patients require rapid and accurate diagnosis since symptoms of the disease are not highly specific and can mimic the symptoms of other neurological disorders. Thus, developing prognostic and diagnostic biomarkers for ALS is urgently needed for effective and rapid diagnosis of the disease [11,12]. We have used SOD1G93A rats in this study since our transgenic rat model expresses human mutant SOD1G93A protein. On the other hand, a rat was chosen as a model because of the high blood volume and serum required to evaluate a wide range of biomarkers, including hematological, hormone, trace elements, minerals, and serum biochemistry.”

Comment

Please change the reference with numbers. i.e.  Fang et al;  Chiò et al.

Answer

We have corrected all references cited in the manuscript.

Comment

Line 254, I think that the sentence “Increased Ca levels in ALS patients have been found” refers to a cytoplasmic Ca increase. https://doi.org/10.1016/j.neuro.2022.01.003

Answer

Ca levels in the mitochondria, serum and cytosol are impaired in the ALS progression in both animal models and humans. We aimed to mention serum Ca levels, there we have added a new article currently published.

Comment

There are also other parameters that may have an importance also considering the severe skeletal muscle involvement (Lanznaster et al. Biomedicines. 2022 Jun 2;10(6):1307. doi: 10.3390/biomedicines10061307). 

Answer

Thank you very much for your precious contributions, we have mentioned indicated biomarkers in our manuscript and added abovementioned reference as written below. However, indicated biomarkers were evaluated only at the early stages of the ALS, we aimed to discuss possible diagnostic biomarkers at the pre-symptomatic and symptomatic stages. Also indicated biomarkers in the manuscript such as SOD3, glutaredoxin 2 (GLRX2) and mitochondrial complex II refer impaired oxidative stress metabolism in the muscle tissue and correlated with the impaired energy metabolism.

“Creatinine has been characterized as an independent prognostic biomarker for ALS and elevated in the earlier phases of the disease as muscle mass destruction and decreased at the late stages of the disease as a hallmark of malnutrition. Additionally, reduced creatinine levels are associated with reduced survival time and poor prognosis in male and female patients [23]. Furthermore, increased activity of SOD3, glutaredoxin 2 (GLRX2) and mitochondrial complex II have been found in the muscle tissue of the ALS patients at the early stages indicating impaired energy metabolism in muscles [24]. We have found, that creatinine levels increased in the SOD1G93A rats of group 0 and decreased compared to the SOD1WT rays by aging and disease progression in groups A, B, C, and D (Fig. 2), indicating creatinine could be used as a diagnostic and prognostic biomarker in ALS.”

Comment

Finally, it should be introduced in the text the reason of the choose of the transgenic rat. The rat model can be close to human pathology? There are differences with respect to the mouse model?

Answer

Dear Reviewer, thank you very much for your comments. We have chosen transgenic rat model because our rats are expressing human mutant SOD1G93A protein. Hemizygous rats express SOD1G93A in the spinal cord ~8-fold above endogenous SOD1. SOD1G93A is also expressed across many brain regions as well as peripheral tissues. By end stage, mutant SOD1 levels accumulate ~16 fold over endogenous, representing a further 2-fold increase in SOD1G93A compared with levels in young, pre-symptomatic rats (6 weeks old). We have conducted several studies in the peripheral tissues of ALS rats, therefore SOD1 expression in the peripheral tissue was crucial for us, also mutant SOD1 expression increased by age and disease progression, so we observed ALS progression correlated with the accumulation of SOD1 and oxidative stress metabolism in the peripheral tissue doi: 10.3389/fbioe.2022.810243.

On the other hand, we have evaluated wide range of biomarkers including serum biochemistry, blood, trace elements, minerals and hormone, therefore we needed high volume of blood and chosen rat model over mice. Our rat model has Also, we have collected tissues to evaluate histopathological, biomechanical and biochemical changes in ALS progression, thus we needed larger tissues. In conclusion, rat model expressing human mutant SOD1G93A protein was suitable for our study. When using mice as animal model, we can take maximum 500-900 µl of blood depending on the age and body weight that cannot be enough to for evaluation of the wide range of the biomarkers as we did. We have taken 5-6 ml blood from the rat heart especially in the groups B, C and D. We have added an explanation in the discussion part as written below according to your comments.

“2.2. Animal Studies

Six male and six female SOD1G93A mutated albino rats were purchased from Taconic with catalog number NTac: SD-Tg(SOD1G93A)L26H. Animals were expressing human mutant SOD1G93A protein as 8-fold more endogenous at the pre-symptomatic stages and 16-fold more at the end stage of the disease (http://www.taconic.com/2148).”

“Discussion

ALS patients require rapid and accurate diagnosis since symptoms of the disease are not highly specific and can mimic the symptoms of other neurological disorders. Thus, developing prognostic and diagnostic biomarkers for ALS is urgently needed for effective and rapid diagnosis of the disease [11,12]. We have used SOD1G93A rats in this study since our transgenic rat model expresses human mutant SOD1G93A protein. On the other hand, a rat was chosen as a model because of the high blood volume and serum required to evaluate a wide range of biomarkers, including hematological, hormone, trace elements, minerals, and serum biochemistry.”

Reviewer 2 Report

This is an interesting and innovative paper. The authors investigated male albino rats divided into 10 groups based on age. Each group was divided into 2 subgroups according to their mutation status  (wild type or SOD 1G 93A). Several hematological and serum biochemical parameters were changed in mutant animals. which may to identify rapid and effective prognostic biomarkers of ALS.

Question:

You used male animals. This is understandable. since ALS is 20% more common in men than in women.  However, with increasing age the incidence of ALS is more equal betwen men and women. So my question is:

Did you ever studied female rats? If yes, were the changes in SOD mutants also independent of age?

Author Response

Dear Reviewer,

We would like to thank you for your precious comments. In this study, we have used both female and male rats, but there was no significant change in biochemical, histopathological, or biomechanical aspects between two genders. In the figure.1. you can find the female rats weights that are significantly reduced in the SOD1G93A rats compared to the SOD1WT rats of groups C and D same as male rats. We have observed same age-independent changes in the SOD1 mutated female rats compared to the wild-type rats same as male rats.

Figure.1. Animal weights of the female SOD1G93A and SOD1WT rats

We have evaluated serum biochemistry and blood parameters in female rats as well, however there was no significant difference between male and female rats. We are currently conducting different studies with female tissue regarding reproductive system and fat metabolism. On the other hand, we have published a paper about the impact of the ALS on the lung tissue correlated with the biomechanical and biochemical changes (doi: 10.3389/fbioe.2022.810243). We have only used male data however we have studied both female and male tissues, however there was no significant changes between two genders as you can see below in the Figure.2, 3 and 4. That means we have observed the same age-independent changes in the SOD1 rats in both genders.

Figure.2. Lung stiffness of the SOD1G93A and SOD1WT rats (Unpublished data)

Figure.3. Lung force of the SOD1G93A and SOD1WT rats (unpublished data)

Figure.4. Lung displacement of the SOD1G93A and SOD1WT rats (unpublished data)

Round 2

Reviewer 1 Report

The authors have satisfactorily answered to my comments